# l-Ornithine-N5-monooxygenase (PvdA) Substrate Analogue Inhibitors for *Pseudomonas aeruginosa* Infections Treatment: Drug Repurposing Computational Studies

**DOI:** 10.3390/biom12070887

**Published:** 2022-06-25

**Authors:** Joseph Christina Rosy, Ewa Babkiewicz, Piotr Maszczyk, Piotr Mrówka, Banoth Karan Kumar, Sankaranarayanan Murugesan, Selvaraj Kunjiappan, Krishnan Sundar

**Affiliations:** 1Department of Biotechnology, School of Bio and Chemical Engineering, Kalasalingam Academy of Research and Education, Krishnankoil 626126, India; christinarosy.j@gmail.com (J.C.R.); selvaraj.k@klu.ac.in (S.K.); 2Department of Hydrobiology, Faculty of Biology, University of Warsaw, 02-089 Warsaw, Poland; e.babkiewicz@biol.uw.edu.pl (E.B.); p.maszczyk@uw.edu.pl (P.M.); 3Department of Biophysics, Physiology and Pathophysiology, Medical University of Warsaw, 5 Chalubinskiego Street, 02-004 Warsaw, Poland; pmrowka@ihit.waw.pl; 4Department of Experimental Hematology, Institute of Hematology and Transfusion Medicine, 5 Chocimska Street, 00-791 Warsaw, Poland; 5Medicinal Chemistry Research Laboratory, Department of Pharmacy, Birla Institute of Technology and Science-Pilani, Pilani 333031, India; p20180042@pilani.bits-pilani.ac.in (B.K.K.); murugesan@pilani.bits-pilani.ac.in (S.M.)

**Keywords:** siderophore, pyoverdine, PvdA, substrate analogues, N-2-succinyl ornithine, cilazapril

## Abstract

*Pseudomonas aeruginosa* is an opportunistic pathogen that can cause acute and severe infections. Increasing resistance to antibiotics has given rise to the urgent need for an alternative antimicrobial agent. A promising strategy is the inhibition of iron sequestration in the bacteria. The current work aimed to screen for inhibitors of pyoverdine-mediated iron sequestration in *P. aeruginosa*. As a drug target, we choose l-ornithine-N5-monooxygenase (PvdA), an enzyme involved in the biosynthesis of pyoverdine that catalyzes the FAD-dependent hydroxylation of the side chain amine of ornithine. As drug repurposing is a fast and cost-efficient way of discovering new applications for known drugs, the approach may help to solve emerging clinical problems. In this study, we use data about molecules from drug banks for screening. A total of 15 drugs that are similar in structure to l-ornithine, the substrate of PvdA, and 30 drugs that are sub-structures of l-ornithine were virtually docked against PvdA. N-2-succinyl ornithine and cilazapril were found to be the top binders with a binding energy of −12.8 and −9.1 kcal mol^−1^, respectively. As the drug-likeness and ADME properties of the drugs were also found to be promising, molecular dynamics studies were performed to further confirm the stability of the complexes. The results of this in silico study indicate that N-2-succinyl ornithine could potentially be explored as a drug for the treatment of *P. aeruginosa* infections.

## 1. Introduction

In the never-ending battle of human hosts and pathogens, the winner is the one who evolves faster. Bacteria, with their relatively small genome and fast proliferation rate, plus plasmid-related genetic plasticity, may evolve faster, particularly with traits that are effective in allowing them to adapt to the environment and increase their ability to infect the host. A game changer in the treatment of microbial infections was the introduction of antibiotics to therapy. Unfortunately, antibiotic resistance is one feature that bacteria use for infection and survival in the human host. Thus, there is an urgent need to find an alternative to antibiotics that has a mechanism of action that is harder for bacteria to overcome [1,2,3]. Targeting iron sequestration is one such alternative strategy allowing the inhibition of pathogenic bacteria proliferation [4,5,6]. 

*Pseudomonas aeruginosa* is a Gram-negative bacterium that causes a variety of infections in humans. It is an opportunistic pathogen that mainly affects immunocompromised individuals, such as HIV positive or cystic fibrosis patients. It is also the major causative agent of complicated urinary tract infections (UTIs). It is estimated that *P. aeruginosa* is responsible for 35% of the reported cases of Catheter Associated Urinary Tract Infections (CAUTIs). This bacterium has been shown to have a high level of intrinsic resistance to many antibiotics. The mechanisms of the resistance are: limiting outer membrane permeability, the induction of efflux systems that pump antibiotics out of the cell, and producing antibiotic-inactivating enzymes, such as β-lactamases [7]. Hence, it is essential to find novel antimicrobial agents that impose weaker evolutionary burden on this bacterium. 

*P. aeruginosa* uses two siderophores for iron sequestration: pyochelin and pyoverdine. Pyoverdine is a fluorescent siderophore which has been shown to have higher affinity to iron than pyochelin [8,9]. It is involved not only in iron sequestration, but also in many other cellular activities. Pyoverdine is essential for the pathogenesis of *P. aeruginosa* and is considered one of the virulence factors of this bacterium [10,11,12,13]. Iron saturated pyoverdine stimulates the release of PvdS, which in turn promotes the expression of two secreted toxins, namely, ToxA and PrpL [12,14,15]. For these reasons, pyoverdine is an attractive therapeutic target used to control *Pseudomonas* infections [16]. 

Substrate analog inhibitors are a class of antimicrobials that are successfully used for treating many infections and disorders. Though a few studies are available on searching for substrate analogues in vitro, to the best of our knowledge, no attempts have been made to find substrate analog inhibitors using an in silico approach. Hence, we believe our studies will lay the groundwork for exploring a novel method of designing substrate analogs in silico. 

We chose l-ornithine-N5-monooxygenase (PvdA), an enzyme involved in the synthesis of pyoverdine, as the drug target in the current studies [17]. The PvdA structure has three domains, namely, the ornithine, NADPH, and FAD-binding domain. Our approach was to find a structural analog for l-ornithine that is the substrate of PvdA among existing drugs. Drug repurposing is a strategy for discovering new applications for approved or investigational drugs that are beyond the original scope of the medical indication. Moreover, new applications can be found for drugs that, despite positive initial studies, did not find their place in market or were withdrawn from the market for different reasons. As existing medicines have already completed safety studies in humans and do not require Phase 1 clinical trials, drug repurposing holds considerable advantages over conventional drug discovery pathways, including significant time saving and research and development costs reduction. 

To achieve the goal of selecting drugs that could serve as substrate analog inhibitors of PvdA, we searched the DrugBank5.0 database for structural analogues of l-ornithine and compounds that include ornithine as their substructures, and we performed molecular docking studies on the selected drugs to identify the top binders of PvdA, potentially the best inhibitors.

## 2. Experimental Section

### 2.1. Collection of Substrate Analogues

Ornithine analogs were collected from a drug bank (www.go.drugbank.com, accessed on 30 November 2020) [18] using a similarity search and substructure search. A total of 15 and 30 drugs were obtained from the similarity and substructure search, respectively. The names and structure of all these drugs are provided in Table 1 and Table 2.

### 2.2. Computational Details

All computations were performed through Debain Linux Operating System running on a computer that has 8 GB RAM and Intel^®^ Core™ i5-4440 CPU (Acer, India), which runs at 3.10 GHz. For molecular docking, AutoDock Vina installed in Debain Linux was used [19]. For the analysis of docking results, the Accelrys^®^ Discovery studio visualizer (Accelrys Software Inc., San Diego, CA, USA) that runs on the Windows operating system was used. Molecular Dynamics studies were performed with Desmond^®^ using the user interface of Schrodinger (Schrodinger Inc., New York, NY, USA).

### 2.3. Preparation of Protein and Ligand Structures

The crystal structure of PvdA was downloaded from Protein Data Bank (PDB; http://www.rcsb.org/pdb/, accessed on 30 November 2020) [20,21]. Detailed information regarding the protein structure is given in Table 3. 

Since the crystal structure is a homodimer, one of the chains was selected for molecular docking studies. The other chain was deleted and hydrogen atoms were added to the protein structure. Water molecules and the heteroatoms, including the FAD and NADPH present in their respective domains, were removed [22]. Charges were added to the protein structure by applying a force field using AutoDock Tools [23]. All the 45 compounds were also prepared for docking using AutoDock Tools. The crystal models provide the enzyme structure at a pH near to neutral, and the docking studies were performed with the same protonation states as per the crystal structure (i.e., at pH 7). Since the optimum pH for the activity of PvdA was reported to be in the range of 8 to 8.5 [24], to understand the effect of pH on the binding of the ligand to PvdA, the protonation state of the charged amino acids of PvdA were changed at pH 8.3 using the H++ server. Molecular docking studies were also performed with this protonated PvdA and ligands.

### 2.4. Molecular Docking Studies

Using AutoDock Tools, a 3D grid box was created in which the protein structure was embedded. Grid parameters were noted for running AutoDock Vina. The binding energies for each conformation of the ligand with the proteins were determined by running AutoDock Vina. Binding energies of each conformation of docked compounds were noted, and the best conformation was chosen based on the binding energy and number of hydrogen bonds that they form with the protein. Analysis of docking was performed by using Discovery Studio Client 4.0. Various types of interactions between ligand and receptor, such as hydrogen bonds, hydrophobic interactions, Van der Waals and electrostatic interactions, were visualized. Along with compounds, the substrate l-Ornithine was also docked with PvdA to serve as a reference.

### 2.5. Prediction of Drug Properties

Absorption, distribution, metabolism, and excretion (ADME) properties of the top binders were predicted using the SwissADME online server [25]. Properties such as molecular weight, number of H-bond donors, number of H-bond acceptors, number of rotatable bonds, LogP, gastrointestinal (GI) absorption, blood brain barrier (BBB) permeation, and bioavailability were also predicted using the same server.

### 2.6. Drug-Likeness

‘Drug-likeness’ is the ability of a molecule to become an oral drug with respect to bioavailability. ‘Drug-likeness’ was predicted by SwissADME using rules as described before [26,27,28,29,30]. These rules as drug-likeness filters often originate from analyses by major pharmaceutical companies aiming to improve the quality of their proprietary chemical collections. The Lipinski filter is the pioneer rule, known as the rule-of-five, which is used by Pfizer. Other filters are: Ghose by Amgen, Veber by GSK, Egan by Pharmacia, and Muegge by Bayer [25]. The details of all the five filters are presented in Table 4. The compounds satisfying these rules are considered likely to be effectively developed into an oral drug [31].

### 2.7. Molecular Dynamics Simulation

Molecular Dynamics (MD) simulation studies help envisage the protein–ligand complex’s actions at the target’s binding site region in the physiological conditions. MD was performed using the Desmond module of Schrodinger developed by the D.E. Shaw research group (academic license) through the system’s builder panel; the orthorhombic simulation box was prepared with the Simple Point–Charge (SPC) explicit water model in such a way that the minimum distance between the protein surface and the solvent surface was 10 Å. Three runs of MD simulations were performed in order to confirm the consistency of the results. Protein–ligand docked complexes were solvated using the orthorhombic SPC water model [32]. The solvated system was neutralized with counter ions, and the physiological salt concentration was limited to 0.15 M. The receptor–ligand complex system was designated with the OPLS AA force field [33]. 

The Reversible reference System Propagator Algorithms (RESPA) integrator, Nose-Hoover chain thermostat, and Martyna-Tobias-Klein barostat were used with 2 ps relaxation time [34,35]. The equilibrated system was used for the final production of MD simulation. The production of the MD simulation was run for 100 ns at 300 K temperatures at 1.0 bar pressure with NPT (Isothermal-Isobaric ensemble, constant temperature, constant pressure, and constant number of particles) ensemble [36] with the default settings of relaxation before simulation.

The MD simulation was run using the MD simulation tool with simulation time setup to 100 ns. Further, for viewing the trajectories and creating a movie, the _out. CMS file was imported, and the movie was exported with high resolution (1280 × 1024) with improved quality. During the MD simulation, the trajectory was written with 1000 frames. To understand the stability of the complex during MD simulation, the protein backbone frames were aligned to the backbone of the initial frame. Finally, the analysis of the simulation interaction diagram was achieved after loading the _outacts file and selected Root Mean Square Deviation (RMSD) and Root Mean Square Fluctuation (RMSF) in the analysis type to obtain the mentioned plots [37]. 

## 3. Results

### 3.1. Structure of PvdA

PvdA is considered a member of the class B flavoprotein monooxygenases, which utilizes FAD as a co-factor and NADPH as the electron donor. The crystal structure is a homodimer; each monomer contains three domains, namely: FAD-binding domain, NADPH-binding domain and Ornithine (substrate) binding domain. These have FAD, NADPH, and N~5~hydroxy-l-ornithine (product) as co-crystallized ligands (Figure 1) [21].

### 3.2. Top Binders of PvdA

The 15 drugs that are similar in structure to l-ornithine and 30 drugs that are substructures of l-ornithine were docked against PvdA, and the binding energies are listed in Table 1 and Table 2, respectively. The substrate of l-ornithine was also docked against PvdA, and its binding energy was found to be −4.9 kcal mol^−1^. Compounds showing the binding energies of less than or equal to −9 kcal.mol^−1^ were designated as top binders. Out of the 30 substructure drugs, N-2-succinyl ornithine, trypanothione, talotrexin, davunetide, CTT-1057, and Cilazapril were found to be the top binders with binding energies of −12.8, −11.7, −11.0, −11.0, −9.4 and −9.1 kcal mol^−1^, respectively. N-2-succinyl-ornithine was found to bind with PvdA with higher affinity, which is evident from the binding energy of −12.8 kcal mol^−1^. It was much higher than the binding energy of the substrate l-Ornithine (−7.7 kcal mol^−1^). Out of the 15 similar drugs docked, none were found to have a binding energy of less than or equal to −9 kcal mol^−1^. 

The top binder N-2-succinyl-ornithine was found to interact with the amino acids SER21, ASN22, TRP52, SER167, PRO168, ARG357, GLY396, and SER415, out of which ARG357 was found to be interacting with three different amino acids of PvdA. The interactions of N-2-succinyl-ornithine with PvdA is shown in Figure 2a, and the 2D interaction diagram is shown in Figure 2b.

The protonation state of the protein was changed according to pH 8.3 and was docked against the ligands. The results indicate that N-2-succinyl ornithine is one of the top binders (Appendix A). Moreover, to evaluate the effect of the presence of cofactor FAD and electron donor NADP+ in the binding of the ligand, a docking study was performed without removing the cofactor and electron donor. In this condition, N-2-succinyl ornithine showed a binding energy of −5.7 kcal mol^−1^ without FAD and NADP+ at pH 8.3. The binding energy was found to be −6.3 kcal mol^−1^ with the cofactor and electron donor in the structure at pH 8.3, which is close to the binding energy obtained in the docking study without NADP+ and FAD at pH 8.3.

This indicates that NADP+ and FAD do not interfere with the binding of the ligands. This claim was further confirmed by docking another ligand, Argininosuccinate, to the protein without NADP+ and FAD. The binding energy in the presence of the cofactors is −6.7 kcal mol^−1^ and without the cofactors is −6.9 kcal mol^−1^.

### 3.3. ADME Properties and Drug-Likeness

The ADME properties of the top binders are listed in Table 5. N-2-succinyl-ornithine and Cilazapril were found to have good ADME properties. Both of these drugs were predicted to possess a high possibility of gastro-intestinal absorption, and they also have a molecular weight of less than 500 g mol^−1^, which is desirable for a drug to work better in vivo. 

The ‘drug-likeness’ of the top binders was also analyzed according to five rules (Table 6). Cilazapril was found to have a greater ‘drug-likeness’, as it has no violations in all the five rules. As Cilazapril is a drug which is currently available on the market, it can easily be tested as a candidate drug without any additional steps. However, since cilazapril is an ACE inhibitor, its side effects should be further evaluated using in vivo studies. The top binder N-2-succinyl ornithine also demonstrates good ‘drug-likeness’ properties, despite the two violations found. This drug has a higher bioavailability score than other drugs and has the potential to be developed into a drug that inhibits PvdA. Hence, N-2-succinyl ornithine was chosen for further molecular dynamics studies.

### 3.4. Molecular Dynamics Simulation

MD simulations were carried out with the PvdA and the top binder N-2-succinyl ornithine to determine the stability of the protein ligand complex and to find the key residues of interaction. Three runs of MD simulations were performed, each of which was run for 100 ns, and the Root Mean Square Deviation (RMSD), Root Mean Square Fluctuation (RMSF), and contacts between proteins and ligands were calculated. The results of one of the three runs are discussed here, and the results of the other runs are provided in the Appendix A. The protein and ligand RMSD shown in Figure 3 (Appendix A) clearly shows that the RMSD does not exceed 3Å for the protein and ligand, which indicates that the complex is stable. The RMSD of the ligand is almost similar to the protein, which confirms the ligand is stably in complex with the protein.

The Ligand Root Mean Square Fluctuation (L-RMSF) is useful for characterizing changes in the ligand atom positions. The average change in the position of all the 13 atoms of the ligand is shown in the plot (Figure 4, Appendix A). This indicates all the atoms underwent minimal or no changes during the 100 ns dynamic simulation. This confirms the stability of the ligand binding to the protein. 

The protein ligand interactions were monitored throughout the simulation, which is provided in Figure 5. The interaction fraction was calculated for each residue interacting with the ligand. The ‘interaction fraction’ is the number of times each residue of PvdA interacts with the ligand. The interactions are normalized over the course of the simulation, 100 ns being 100%. For example, a value of 0.6 indicates that for 60% of the total simulation time, the particular residue was interacting with the ligand. Values above 1.0 are possible, as some of the residues have more than one interaction with the ligand. ARG357 was found to be the key residue of interaction, as it has three interactions, namely H-bonding, and ionic and water-bridge with the atoms of N-2-succinyl-ornithine (Figure 2b, Figure 5, Appendix A). The number of times the ligand contacts the protein through the 100 ns simulation is also shown in Figure 6, Appendix A. This contact diagram strengthens our finding that Arg357 is important for binding, as it makes more than one contact with the ligand.

## 4. Discussion

Substrate analogs, via the structural resemblance, competitively inhibit the enzyme activity occupying the same active site where the substrate binds. Few attempts have been made to rationally design substrate analog inhibitors as drugs to treat infections [38,39,40]. The confirmation that this is a fruitful approach lies in the presence of antimetabolite drugs in the pharmaceutical market that are analogues of substrates for microbial, viral infected, or tumor cell key enzymes. A well-known example is methotrexate, a folic acid analog, inhibitor of dihydrofolate reductase (DHFR), which is an enzyme that participates in the tetrahydrofolate synthesis used in cancer chemotherapy and to treat autoimmune diseases [41]. 

Though substrate analog inhibitors have been used for many decades, there were no reports found in the literature for the in silico rational design of substrate analog inhibitors. The current study’s direct aim was to find substrate analogs of the drug target, PvdA, an enzyme that converts l-ornithine into N~5~hydroxy-l-ornithine, but we also wanted to explore the usefulness of in silico methods for designing rational inhibitors against an enzyme. We believe our results prove the significant potential of computational analysis combining data including structures, safety, involvement in metabolic pathways, etc., of existing drugs in order to find new possible applications and/or interactions. Using a set of computer programs and available data bases (DrugBank, Protein Data Bank), we selected two strong candidates to become antimicrobial agents for *P. aeruginosa* control: N~2~-Succinylornithine and Cilazapril.

N~2~-Succinylornithine was found to bind to the target enzyme with an energy of −12.8 kcal mol^−1^. The molecular dynamics studies clearly indicate that this molecule forms a stable complex with PvdA and ARG357 is the key residue of the interaction. The ADME and drug-likeness properties of N-2-Succinylornithine were also found to be promising. The compound is reported to be found naturally in *Trypanosoma brucei* and to play a role as an *Escherichia coli* metabolism, but to date it has not been used as a drug [DrugBank and PubChem]. Its possible role in *P. aeruginosa* metabolism has been suggested before [42], which in our opinion, confirms the usefulness of the methodology used in this study.

The second selected drug found to interact with PvdA with a good binding affinity was cilazapril. Additionally, the compound demonstrated better ADME and drug likeness properties. Cilazapril is a commercially available angiotensin-converting enzyme inhibitor (ACE inhibitor; ACEI) used for the treatment of high blood pressure and heart failure. [43]. Its brand names are Dynorm, Inhibace, Vascace, and others in different countries. It has a relatively good safety profile, although some side effects have been reported [44]. Although ACEI are not neutral to human, we think it is possible to find a therapeutic window for microbial treatment or the drug could be used in external applications, in the form of ointments, aerosol gels, and as a component of ready-made dressings. The use of cilazapril or any ACEI in the treatment of *P. aeruginosa* infections has not been reported before, neither has any information of its influence on the bacteria.

Although we only demonstrate an in silico analysis and selection algorithm, we believe our results provide a strong rational to test the selected compounds vs *P. aeruginosa*. The natural next step is to perform in vitro studies that will confirm or disprove our results. Regardless of that, our research shows new possibilities of searching for drugs, and we hope that they will popularize drug repurposing as an effective method of searching for candidates for in vitro and in vivo tests, helping to solve many clinical problems.

## 5. Conclusions

Our study opens a new direction of fast and low-cost screening for promising clinically effective molecules. The selected molecules—N-2-Succinylornithine and cilazapril—should be further studied and possibly developed into substrate analog inhibitors of PvdA. Further studies are required to prove the efficacy of these molecules to inhibit PvdA activity and their usefulness in *P. aeruginosa* infections treatment. Moreover, we show a full and complex computational approach for drug repurposing.

## Figures and Tables

**Figure 1 biomolecules-12-00887-f001:**
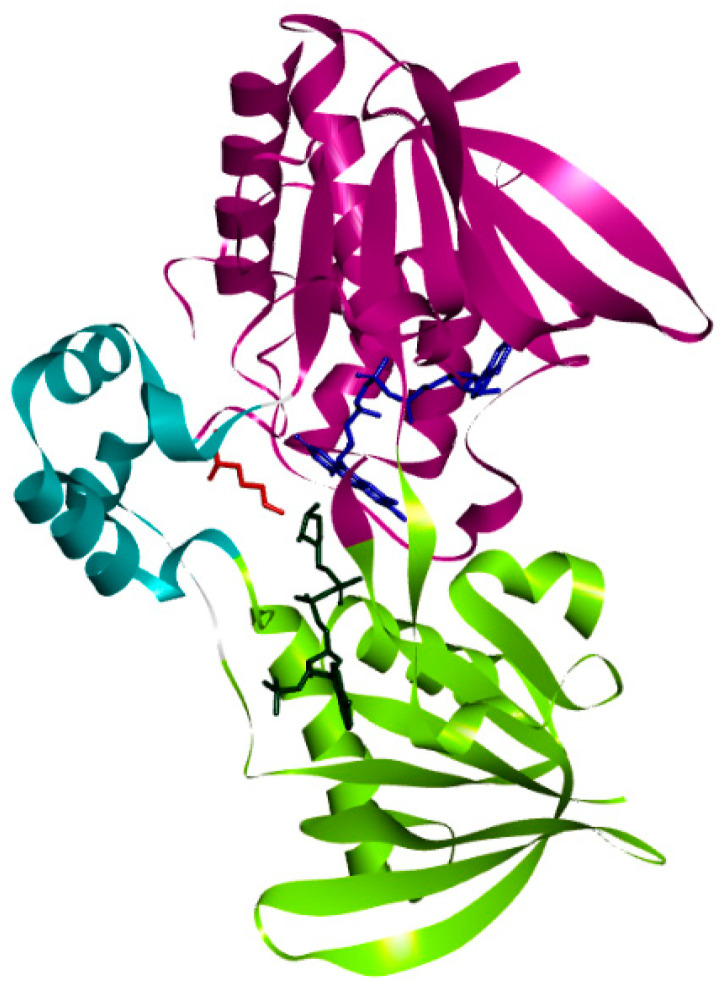
Structure of PvdA: The three domains, namely: the Ornithine-binding domain (cyan), FAD-binding domain (purple), and NADPH-binding domain (green) present in the crystal structure of PvdA are shown.

**Figure 2 biomolecules-12-00887-f002:**
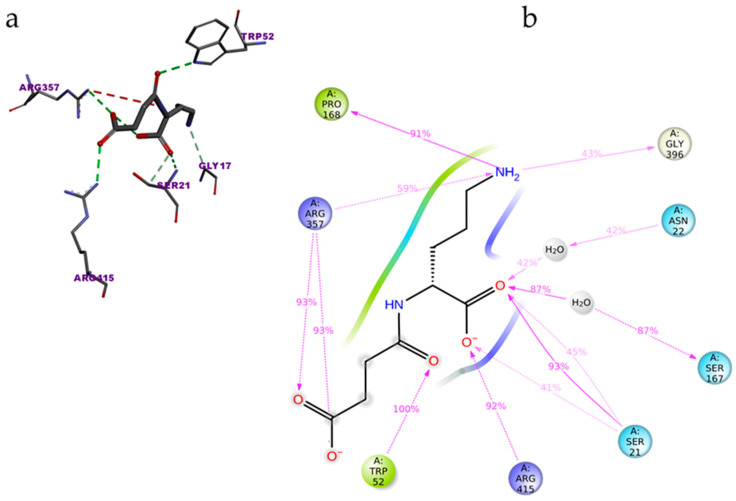
Interaction of N-2-Succinyl-ornithine with PvdA: amino acids of PvdA interacting with the top binder N-2-succinyl-ornithine (**a**); and 2D-interaction plot of N2-Succinyl-ornithine interacting with PvdA. ARG357 was found to make three interactions with PvdA (**b**).

**Figure 3 biomolecules-12-00887-f003:**
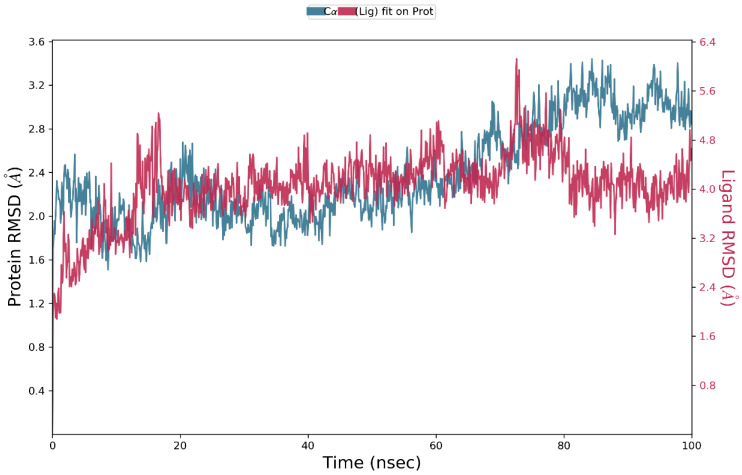
RMSD of protein and ligand throughout the 100 ns simulation; the interactions of PvdA and N-2-succinyl-ornithine were found to be stable after 40 ns.

**Figure 4 biomolecules-12-00887-f004:**
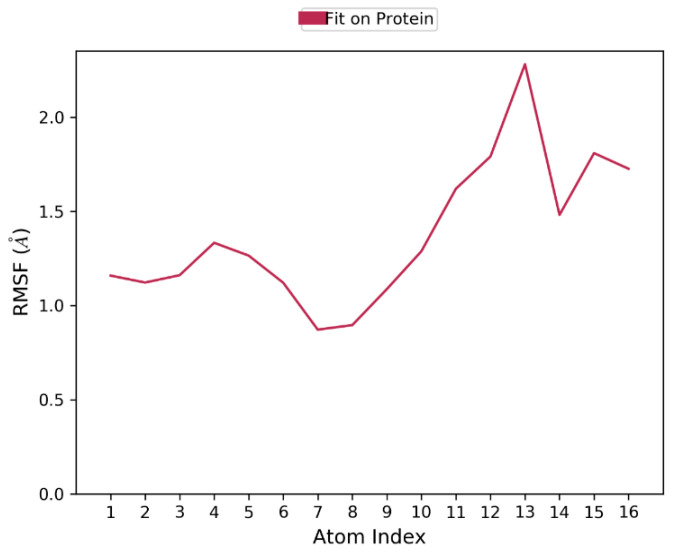
Ligand RMSF plot for characterizing changes in the ligand atom positions. The RMSF of all the atoms are not more than 2Å, which indicates that the binding is stable.

**Figure 5 biomolecules-12-00887-f005:**
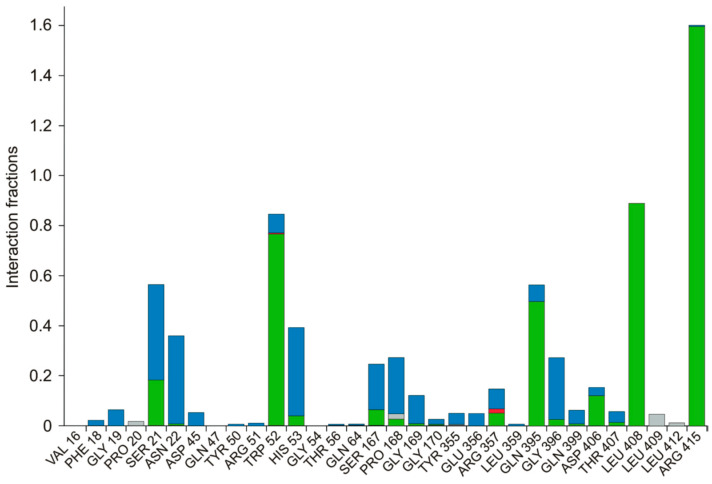
Various interactions of the ligand with the amino acids are represented. Amino acids of PvdA that have interactions with the ligand and the type of interaction are provided. Green—Hydrogen Bond; Red—Ionic; Lavender—Hydrophobic; Blue—Water Bridge. Arg357 was found to have three kinds of interactions, namely, a hydrogen bond, water-bridge, and ionic interaction. The interaction fraction is the number of interactions had by a residue normalized by the simulation time during which the interaction existed. An interaction fraction that is more than 1 is possible when the particular residue has more than one interaction.

**Figure 6 biomolecules-12-00887-f006:**
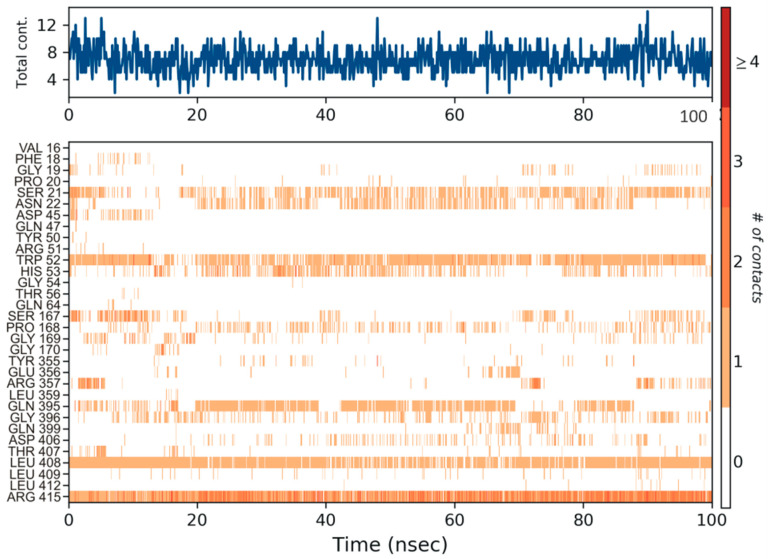
The number of contacts made by the ligand throughout the 100 ns simulation is shown in the top panel (appears in blue). Some residues make more than one specific contact with the ligand, which is represented by a darker shade of orange in the bottom panel, according to the scale to the right of the plot. Arg357 was found to make more than one single contact with the ligand.

**Table 1 biomolecules-12-00887-t001:** Structures and binding energies of drugs that are structurally similar to l-Ornithine.

S. No.	Name of the Drug	Structure	Binding Energy (kcal mol^−1^)
1.	(2s)-2,8-Diaminooctanoic Acid	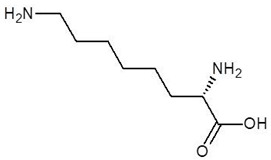	−5.2
2.	2,6-Diaminopimelic Acid	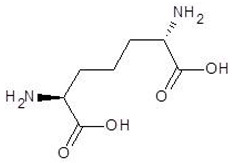	−5.1
3.	2-Methylleucine	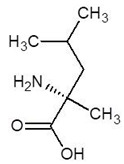	−4.9
4.	Allo-Isoleucine	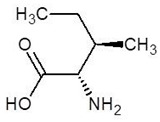	−4.8
5.	4-Carboxy-4-Aminobutanal	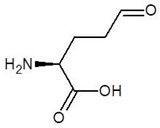	−4.6
6.	d-Leucine	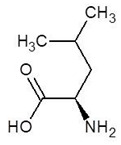	−4.6
7.	2-Aminopimelic Acid	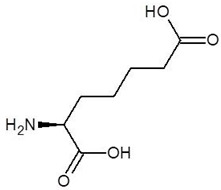	−4.6
8.	d-Lysine	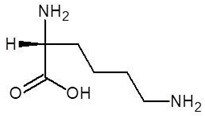	−4.6
9.	2-Amino-6-Oxo-Hexanoic Acid	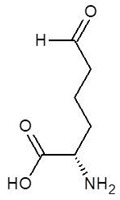	−4.5
10.	d-Glutamine	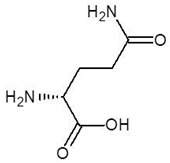	−4.5
11.	6-hydroxy-l-norleucine	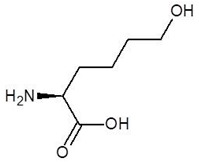	−4.4
12.	Norvaline	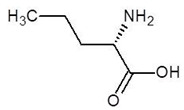	−4.4
13.	5-Hydroxy Norvaline	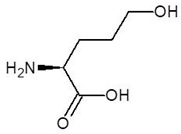	−4.3
14.	Alpha-Aminobutyric Acid	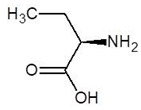	−4.1
15.	Delta-Amino Valeric Acid	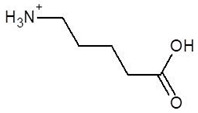	−3.9

**Table 2 biomolecules-12-00887-t002:** Structures and binding energies of drugs that have substructures of l-Ornithine.

S. No.	Name of the Drug	Structure	Binding Energy (kcal mol^−1^)
1.	N-2-Succinyl ornithine	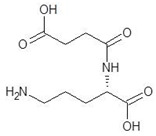	−12.8
2.	Trypanothione	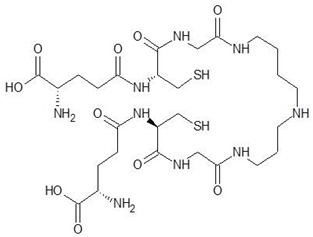	−11.7
3.	Talotrexin	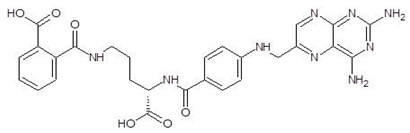	−11.0
4.	Davunetide	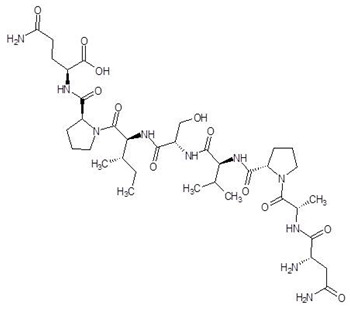	−11.0
5.	CTT-1057	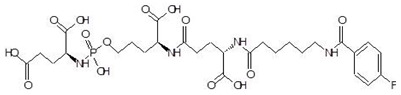	−9.4
6.	Cilazapril	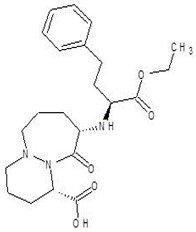	−9.1
7.	S-P-Nitrobenzyloxycarbonylglutathione	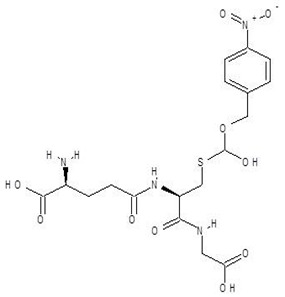	−8.3
8.	Glutathione disulfide	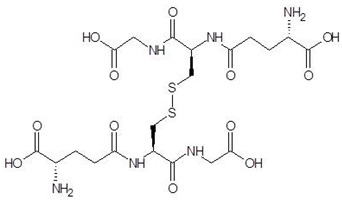	−8.1
9.	S-Hydroxymethyl Glutathione	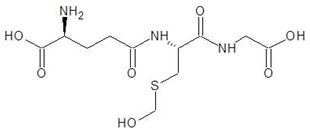	−7.6
10.	Glutathione	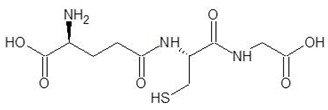	−7.2
11.	Argininosuccinate	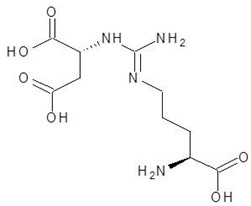	−6.9
12.	Glutathione Sulfinate	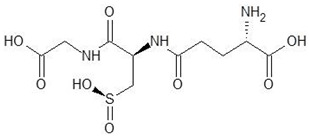	−6.6
13.	N-(Phosphonoacetyl)-l-Ornithine	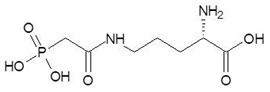	−6.3
14.	N-Alpha-l-Acetyl-Arginine	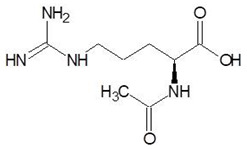	−6.0
15.	Gamma-Glutamylcysteine	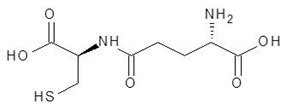	−6.0
16.	N-omega-nitro-l-arginine methyl ester	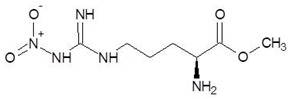	−6.0
17.	S-methyl-glutathione	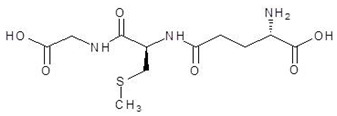	−5.9
18.	N-Acetyl-l-Citrulline	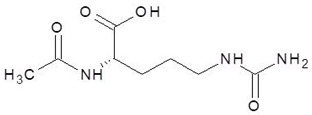	−5.8
19.	Nitroarginine	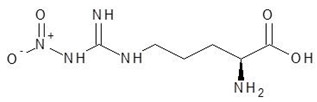	−5.8
20.	N3, N4-Dimethylarginine	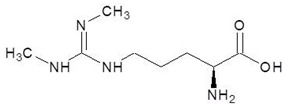	−5.6
21.	l-Eflornithine	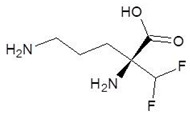	−5.5
22.	5-N-Allyl-Arginine	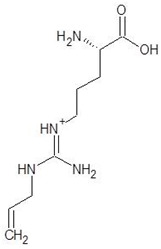	−5.5
23.	Glutamine t-butyl ester	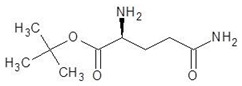	−5.5
24.	N, N-dimethylarginine	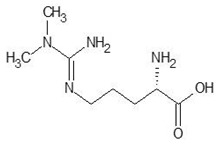	−5.4
25.	Aceglutamide	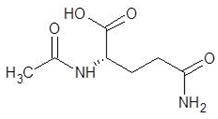	−5.4
26.	Tilarginine	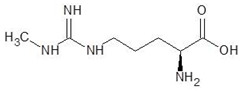	−5.4
27.	l-Citrulline	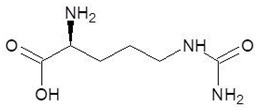	−5.3
28.	Glutamine hydroxamate	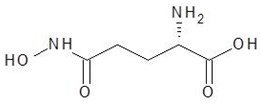	−5.2
29.	N5-Methylglutamine	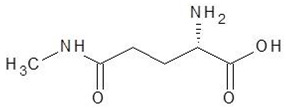	−4.9
30.	l-Thiocitrulline	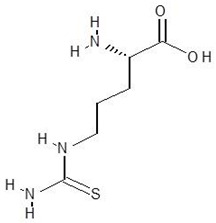	−4.9

**Table 3 biomolecules-12-00887-t003:** Details of the PvdA structure.

Name of the Protein	l-Ornithine-N5-Monooxygenase (or) Ornithine Hydroxylase
Number of amino acids	463
PDB Id	3S5W
Resolution	1.9 Å
Number of chains	2 (Homodimer)
Number of Domains	Three FAD Binding Domain NADPH Binding Domain Ornithine Binding Domain
Amino acids interacting with natural ligand (N5- hydroxyl ornithine)	Lys69, Asn254, Phe257, Asn284, Ser410

**Table 4 biomolecules-12-00887-t004:** Drug-likeness filters used in the study.

Lipinski (Lipinski et al., 1997)	Ghose Ghose, (Viswanadhan, Wendoloski, 1999)	Veber (Veber et al., 2002)	Egan (Egan, Merz, Baldwin, 2000)	Muegge (Muegge, Heald, Brittelli, 2001)
MW ≤ 500 MLOGP ≤ 4.15 N or O ≤ 10 NH or OH ≤ 5	160 ≤ MW ≤ 480 −0.4 ≤ WLOGP ≤ 5.6 40 ≤ MR ≤ 130 20 ≤ atoms ≤ 70	Rotatable bonds ≤ 10 TPS ≤ 140	WLOGP ≤ 5.88 TPSA ≤ 131.6	200 ≤ MW ≤ 600 −2 ≤ XLOGP ≤ 5 TPSA ≤ 150 Num. rings ≤ 7 Num. carbon > 4 Num. heteroatoms > 1 Num. rotatable bonds ≤ 15 H-bond acc. ≤ 10 H-bond don. ≤ 5

**Table 5 biomolecules-12-00887-t005:** ADME properties of top binders.

Name of the Compound	Molecular Weight (g × mol^−1^)	H-Bond Acceptors	H-Bond Donors	Rotatable Bonds	LogP_o/w_	GI Absorption	BBB Permeation	Bioavailability Score
N2- Succinyl-l-ornithine	232.23	6	4	9	−1.18	High	No	0.56
Trypanothione	723.86	13	11	33	−4.20	Low	No	0.17
Talotrexin	573.56	10	7	14	0.18	Low	No	0.11
Davunetide	824.92	13	10	29	−3.07	Low	No	0.17
CTT-1057	706.61	16	9	28	−0.12	Low	No	0.11
Cilazapril	417.50	7	2	9	1.37	High	No	0.55

**Table 6 biomolecules-12-00887-t006:** Drug-likeness of top binders.

Name of the Compound	Lipinski	Ghose	Veber	Egan	Muegge
N-2-Succinyl ornithine	Yes; 0 violation	No; 1 violation: WLOGP < −0.4	Yes; 0 violation	Yes; 0 violation	No; 1 violation: XLOGP3 < −2
Trypanothione	No; 3 violations: MW > 500, N or O > 10, NH or OH > 5	No; 4 violations: MW > 480, WLOGP < −0.4, MR > 130, #atoms > 70	No; 2 violations: Rotors > 10, TPSA > 140	No; 1 violation: TPSA > 131.6	No; 6 violations: MW > 600, XLOGP3 < −2, TPSA > 150, Rotors > 15, H-acc > 10, H-don > 5
Talotrexin	No; 3 violations: MW > 500, N or O > 10, NH or OH > 5	No; 2 violations: MW > 480, MR > 130	No; 2 violations: Rotors > 10, TPSA > 140	No; 1 violation: TPSA > 131.6	No; 2 violations: TPSA > 150, H-don > 5
Davunetide	No; 3 violations: MW > 500, N or O > 10, NH or OH > 5	No; 4 violations: MW > 480, WLOGP < −0.4, MR > 130, #atoms > 70	No; 2 violations: Rotors > 10, TPSA > 140	No; 1 violation: TPSA > 131.6	No; 6 violations: MW > 600, XLOGP3 < −2, TPSA > 150, Rotors > 15, H-acc > 10, H-don > 5
CTT-1057	No; 3 violations: MW > 500, N or O > 10, NH or OH > 5	No; 4 violations: MW > 480, WLOGP < −0.4, MR > 130, #atoms > 70	No; 2 violations: Rotors > 10, TPSA > 140	No; 1 violation: TPSA > 131.6	No; 6 violations : MW > 600, XLOGP3 < −2, TPSA > 150, Rotors > 15, H-acc > 10, H-don > 5
Cilazapril	Yes; 0 violation	Yes; 0 violation	Yes; 0 violation	Yes; 0 violation	Yes; 0 violation

# denotes the number.

## Data Availability

The original contributions presented in the study are included in the article, further inquiries can be directed to the corresponding author.

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
