# Peer review of "l-Ornithine-N5-monooxygenase (PvdA) Substrate Analogue Inhibitors for *Pseudomonas aeruginosa* Infections Treatment: Drug Repurposing Computational Studies"

_biomolecules, 2022, doi:10.3390/biom12070887_

Round 1

Reviewer 1 Report

The manuscript “L-ornithine-N5-monooxygenase (PvdA) substrate analogue inhibitors for Pseudomonas aeruginosa infections treatment: Drug repurposing computational studies” by Joseph Christina Rosy et al. has characterized the inhibitory activities of 45 DrugBank molecules to PvdA using molecular docking and molecular dynamics (MD) simulations in order to repurpose these molecules for P. aeruginosa infections treatment. 2 molecules were found promising, which appears to be a starting point for future in-depth in silico and/or in vitro studies. I think this manuscript merits publication in Biomolecules. But following concerns should be well addressed before it becomes publishable.

1. Major concerns about the molecular docking:

1) PvdA catalyzes the reaction L-ornithine + NADPH + O2 à N5-hydroxy-L-ornithine + NADP+ + H2O, and requires the cofactor FAD [Visca P., et al., J. Bacteriol., 1994, 176 (4): 1128–1140; Ge L., et al., J. Bacteriol., 2006, 188 (20): 7205–7210; Meneely K.M., et al., Biochemistry, 2007, 46 (42): 11930–11937]. The structure (PDB ID: 3S5W) [Olucha J., et al., J. Biol. Chem., 2011, 286 (36): 31789–31798] that the authors used includes L-ornithine, NADP, and FAD. But both NADP and FAD were removed from the docking. The docked ligands thus leave the PDB L-ornithine pocket and bind to the PDB NADP pocket, which can be seen from the interacting pattern displayed in Fig. 2. Thus, I have a very good reason to suspect that in vitro those ligands are competing with NADPH. If NADPH has a higher binding affinity to the pocket, the ligand ranking based on the estimated binding energies (Tables 1 & 2) do not make full sense. Having that this, I think the authors have missed a “control” calculation, i.e., binding energy of NADPH (or NADP+).

2) Note that the pocket investigated in this study has multiple ionizable residues such as Asp45, His53, Arg106, Arg357, and Arg415. Lots of the ligands tabulated in Tables 1 & 2 have ionizable residues, e.g., a top binder N2-succinyl-L-ornithine has two –COOH and one –NH3+ groups. The charge states of a ligand and the receptor residues can have huge impact on the binding affinity [Harris R.C., et al., J. Phys. Chem. Lett., 2017, 8 (19): 4832–4837]. The authors have not clearly stated in the method section how they dealt with the charge states of the ligand and PvdA residues, i.e., at which pH they did the docking (and MD simulations). pH 7 I guess? Please be advised that the optimum pH for PvdA is 8.0–8.5 [Ge L., et al., J. Bacteriol., 2006, 188 (20): 7205–7210; Meneely K.M., et al., Biochemistry, 2007, 46 (42): 11930–11937].

2. About the molecular docking and MD simulations:

1) Did the MD simulations give the same (or highly comparable) binding patterns as those generated by the molecular docking (Fig. 2)? Note that the protein and ligand RMSDs (Figs. 3, S1, S5), ligand RMSF (Figs. 4, S2, S6) cannot give essential binding details because they are way too “macroscopic”. Although Figs. 5, 6, S3, S4, S7, and S8 give those information, but I don’t know what the “interactions fraction” (i.e., y-axes of Figs. 5, S3, and S7) are.

2) My reading of the RMSDs (Figs. 3, S1, S5) is that none of the three independent simulations have converged. Note that 100 ns is “short” for protein–ligand MD simulations. I suggest the authors extend the MD simulations to at least 15–200 ns and update those MD analyses.

3. Other comments:

1) Inconsistent font and font sizes make reading not so pleasant.

2) Some contents in Tables 3 and 6 are not clearly shown.

Author Response

Response to Reviewers Comments

Title of the manuscript: L-ornithine-N5-monooxygenase (PvdA) substrate analogue inhibitors for Pseudomonas aeruginosa infections treatment: Drug repurposing computational studies

Manuscript ID: biomolecules-1770521

We would like to thank the Editors and the Reviewers for the insightful comments that lead to important improvements of the manuscript. Additional analysis suggested by the Reviewers were performed and the results are incorporated into the manuscript. The changes in the manuscript are highlighted in yellow.

Reviewer 1

The manuscript “L-ornithine-N5-monooxygenase (PvdA) substrate analogue inhibitors for Pseudomonas aeruginosa infections treatment: Drug repurposing computational studies” by Joseph Christina Rosy et al. has characterized the inhibitory activities of 45 DrugBank molecules to PvdA using molecular docking and molecular dynamics (MD) simulations in order to repurpose these molecules for P. aeruginosa infections treatment. 2 molecules were found promising, which appears to be a starting point for future in-depth in silico and/or in vitro studies. I think this manuscript merits publication in Biomolecules. But following concerns should be well addressed before it becomes publishable.

Comment 1: PvdA catalyzes the reaction L-ornithine + NADPH + O2 à N5-hydroxy-L-ornithine + NADP+ + H2O, and requires the cofactor FAD [Visca P., et al., J. Bacteriol., 1994, 176 (4): 1128–1140; Ge L., et al., J. Bacteriol., 2006, 188 (20): 7205–7210; Meneely K.M., et al., Biochemistry, 2007, 46 (42): 11930–11937]. The structure (PDB ID: 3S5W) [Olucha J., et al., J. Biol. Chem., 2011, 286 (36): 31789–31798] that the authors used includes L-ornithine, NADP, and FAD. But both NADP and FAD were removed from the docking. The docked ligands thus leave the PDB L-ornithine pocket and bind to the PDB NADP pocket, which can be seen from the interacting pattern displayed in Fig. 2. Thus, I have a very good reason to suspect that in vitro those ligands are competing with NADPH. If NADPH has a higher binding affinity to the pocket, the ligand ranking based on the estimated binding energies (Tables 1 & 2) do not make full sense. Having that this, I think the authors have missed a “control” calculation, i.e., binding energy of NADPH (or NADP+).

Response:

Thank you for the insightful suggestions. Explaining our approach, in fact the molecular docking was first performed without removing the cofactor- FAD and the electron donor NADP+. The protonation state of the protein was changed to pH 8.3 (based on the literature quoted by the Reviewer). The results are shown below and also presented in the supplementary Table S1 in the manuscript. The compounds which are showing binding energies less than -6.1 kcal mol-1 were considered as top binders. N-2 succinyl ornithine was found to be one of the top binders with a binding energy of -6.3 kcal mol-1. These results are in alignment with the previous study, in which N-2-succinyl ornithine was found to be the top binder. Interestingly, all the top binders are amino acid derivatives and that give us a clue that modified amino acids can be developed as potential substrate analogue inhibitors of PvdA.

Docking study was also performed with protein without NADP+ and FAD, in which we found that N2-Succinyl ornithine was found to bind to the protein with a binding energy (-5.7 kcal mol-1) which is not highly different from the binding energy obtained in the docking study with NADP+ and FADP. This indicates that NADP+ and FAD do not interfere with the binding. This claim was further confirmed by docking another top binder, Arginosuccinate to the protein with and without NADP and FAD. The binding energy with cofactors is -6.7 kcal mol-1 and without cofactors is -6.9 kcal mol-1.

For further clarification on whether NADP+ might compete with the ligand, another docking study was performed with NADP+ as ligand. The binding energy was found to be -9.1 kcal mol-1 which is higher than the binding energy of the top binding ligand (N2-succinyl ornithine) which is -12.8 kcal mol-1. This confirms that the top binder has more binding affinity towards the protein than NADP+.

Additionally, it is worth to mention that none of the residues of PvdA that are bound by NADP+ (SER218, SER286, THR353, GLY215 and ARG240) in the crystal structure was found to interact with N2-succinyl ornithine. This can be another argument for that the binding of the ligands (though they bind to NAD binding domain) is not affected by NADP+.

Comment 2: Note that the pocket investigated in this study has multiple ionizable residues such as Asp45, His53, Arg106, Arg357, and Arg415. Lots of the ligands tabulated in Tables 1 & 2 have ionizable residues, e.g., a top binder N2-succinyl-L-ornithine has two –COOH and one –NH3+ groups. The charge states of a ligand and the receptor residues can have huge impact on the binding affinity [Harris R.C., et al., J. Phys. Chem. Lett., 2017, 8 (19): 4832–4837]. The authors have not clearly stated in the method section how they dealt with the charge states of the ligand and PvdA residues, i.e., at which pH they did the docking (and MD simulations). pH 7 I guess? Please be advised that the optimum pH for PvdA is 8.0–8.5 [Ge L., et al., J. Bacteriol., 2006, 188 (20): 7205–7210; Meneely K.M., et al., Biochemistry, 2007, 46 (42): 11930–11937].

Response:

Thank you for pointing out this weakness in the methodology. As suggested, the charge states were added in the methods section. Also, the protonation states of protein and ligand were changed to pH 8.3 using H++ server and these structures were used for docking and MD studies. The results were found to be different from the previous runs and was not in alignment with the docking results. As the physiological pH at which the drug originally intended to act on PvdA will be around neutral, we would like to adopt the MD simulations done previously at pH 7.

Comment 3: About the molecular docking and MD simulations

1) Did the MD simulations give the same (or highly comparable) binding patterns as those generated by the molecular docking (Fig. 2)? Note that the protein and ligand RMSDs (Figs. 3, S1, S5), ligand RMSF (Figs. 4, S2, S6) cannot give essential binding details because they are way too “macroscopic”. Although Figs. 5, 6, S3, S4, S7, and S8 give that information, but I don’t know what the “interactions fraction” (i.e., y-axes of Figs. 5, S3, and S7) are.

Response:

Out of three simulations carried out, two runs were found to be consistent in the results whereas one did not converge to the other two results.

‘Interaction fraction’ is the number of times each residue of PvdA interacts with the ligand. The interactions are normalized over the course of simulation, 100ns being 100%. For example, a value of 0.6 indicates that 60% of the total simulation time the particular residue was interacting with the ligand. Values above 1.0 are possible as some of the residues make more than one interaction with the ligand as in the case of ARG357 (Figure 5). Each interaction is coloured based on the type of interaction (legend provided in the figure) and the interactions are represented as stacked bar.

2) My reading of the RMSDs (Figs. 3, S1, S5) is that none of the three independent simulations have converged. Note that 100 ns is “short” for protein–ligand MD simulations. I suggest the authors extend the MD simulations to at least 15–200 ns and update those MD analyses.

Response:

We agree that extended time of the MD would increase consistency of our results and better simulate the native situation but unfortunately we deal with technical problem to do that. We have tried to extend the simulation for 200 ns. but we are not able to execute it due to too low capacity of the software. We are working on increasing our ability to perform this type of analysis, but it will not be possible in the time necessary to respond to the comments of this publication. We hope that you will find presented results sufficient to confirm the conclusions of our work.

  1. Other comments:

1) Inconsistent font and font sizes make reading not so pleasant.

Response: This has been fixed and the text has uniform font size.

2) Some contents in Tables 3 and 6 are not clearly shown

Response: This has been rectified in the revised manuscript.

Reviewer 2 Report

Manuscript Number: biomolecules-1770521

entitled: L-ornithine-N5-monooxygenase (PvdA) substrate analogue inhibitors for Pseudomonas aeruginosa infections treatment: Drug repurposing computational studies

 This is a well-conducted scientific study, done thoroughly and expressed concisely. Therefore, the manuscript is suitable for Biomolecules after considering the below comments:

1.      Please redraw with higher quality all Tables and Figures. Please paste to Tables figures in one format, possibly in tiff format.

2.      There is a problem with text formatting, and style of presentation (unfortunately insufficient quality).

3.      Would you please add the UIPAC names to trivial names, e.g., in Table 2, there is N-2-Succinyl ornithine, please add (S)-5-amino-2-(3-carboxypropanamido)pentanoic acid.

Author Response

Response to Reviewers Comments

Title of the manuscript: L-ornithine-N5-monooxygenase (PvdA) substrate analogue inhibitors for Pseudomonas aeruginosa infections treatment: Drug repurposing computational studies

Manuscript ID: biomolecules-1770521

We would like to thank the Editors and the Reviewers for the insightful comments that lead to important improvements of the manuscript. Additional analysis suggested by the Reviewers were performed and the results are incorporated into the manuscript. The changes in the manuscript are highlighted in yellow.

Reviewer 2

Reviewer Comments:

This is a well-conducted scientific study, done thoroughly and expressed concisely. Therefore, the manuscript is suitable for Biomolecules after considering the below comments:

Comment 1: Please redraw with higher quality all Tables and Figures. Please paste to Tables figures in one format, possibly in tiff format.

Response: The tables and figures are modified and presented.

Comment 2: There is a problem with text formatting, and style of presentation (unfortunately insufficient quality).

Response: The text formatting is redone as per the journal format.

Comment 3: Would you please add the UIPAC names to trivial names, e.g., in Table 2, there is N-2-Succinyl ornithine, please add (S)-5-amino-2-(3-carboxypropanamido)pentanoic acid.

Response: The Tables 1 and 2 are modified to include both IUPAC names and trivial names.

Round 2

Reviewer 1 Report

In the revised manuscript, the authors have appropriately addressed all my questions and concerns. I believe the manuscript is in good shape and publishable.

Reviewer 2 Report

Manuscript Number: biomolecules-1770521

entitled: L-ornithine-N5-monooxygenase (PvdA) substrate analogue inhibitors for Pseudomonas aeruginosa infections treatment: Drug repurposing computational studies

The authors conducted important data. This is an interesting paper. The present version of the manuscript is very well developed and the manuscript is well written. The data presented are new and relevant. The present version is much better; therefore, the manuscript is suitable for publication in its current form.